# How to Improve Green Innovation Performance: A Conditional Process Analysis

**Na Wang** [1,2,*], **Jinshan Zhang** [1], **Xiue Zhang** [1] and **Wei Wang** [3]

1 Business School, Jilin University, Changchun 130012, China; zhangjs@jlu.edu.cn (J.Z.); zhangxe@jlu.edu.cn (X.Z.)
2 Business School, Changchun Science and Technology University, Changchun 130600, China
3 Training Department, Shenyang Institute of Technology, Shenyang 113122, China; kathyweiwei1110@163.com
* Correspondence: wna18@mails.jlu.edu.cn

**Abstract:** Green innovation strategy (GIS) is an appropriate choice for most enterprises to deal with environmental problems. Organizational green learning (OGL) enables enterprises to obtain more updated environmental knowledge and promote green innovation performance (GIP). It is unclear whether adopting green innovation strategy is inevitably beneficial to green product innovation and green process innovation, and the studies in this area are still incomplete. According to the Natural Resource-Based View and Knowledge-Based View, this study advances a conditional processmodel to understand how green innovation strategy impacts green innovation performance through organizational green learning in a context of green technological turbulence (GTT). We conducted an empirical study with a probabilistic sample of 316 innovative enterprises using the partial least squares and regression analysis in order to verify the research framework. The results show a positive relationship between green innovation strategy and green innovation performance, organizational green learning played a partial mediating effect, and green technology turbulence significantly moderated the relationship between organizational green learning and green innovation performance. The impact of organizational green learning on green innovation performance is greater when green technology turbulence is higher than when it is low. These findings extend the green innovation performance research and practice.

**Keywords:** green innovation strategy; organizational green learning; green technology turbulence; green innovation performance

## 1. Introduction

The environmental issue is one of the major challenges of our time. From shifting weather patterns that threaten food production, to rising sea levels that increase the risk of catastrophic flooding, the impacts of climate change are global in scope and unprecedented in scale. Without drastic action today, adapting to these impacts in the future will be more difficult and costly [1]. It is clear that the role of human influence on the climate system is undisputed and human actions still have the potential to determine the future course of the climate. Strong and sustained reductions in emissions of carbon dioxide and other greenhouse gases should be taken into consideration to limit climate change [2]. As the dual subject of social life and commercial activities, enterprises are not only the "initiator" of environmental problems, but can also offer the "solution" of environmental improvement. The strategy adopted by the enterprise directly affects the change of environment; of course, its implementation is also restricted by the change of environment. Despite debate over whether to be green or non-green [3], enterprises have recognized the need to take measures to avert environmental degradation and balance their social responsibility with economic benefits [4]. Enterprises should choose a path of green, low-carbon and sustainable development. At present, enterprises have realized the importance of environmental problems within the context of the sustainable development of the enterprise.

As a green innovation strategy requires greater costs and rich resources, and holds much uncertainty, many enterprises have not yet incorporated environmental issues into their strategic planningnor implemented green innovation strategies with sustainable development as the principle and goal. Although enterprises in different industries and different enterprises within the industry may differ in green innovation strategies due to various reasons, enterprises viewing green innovation strategies with complete disregard will find it difficult to survive [5].

Green innovation strategy considers both environmental and economic benefits. Similar concepts to green innovation strategy include sustainable innovation strategy, ecological innovation strategy, and environmental innovation strategy. Through comparison, we find that there are only slight differences in the definition and description of the four concepts, which belong to the same theme in their concerned content, so they can be used interchangeably to a large extent [6]. GIS means that enterprises take the initiative to reduce the negative impact on the environment in their business activities and incorporate environmental responsibility into their strategic planning [7]. Enterprises must design and develop more environmentally friendly processes and products to reduce the negative impact on the environment and maintain the sustainable operation of enterprises by incorporating environmental issues into the strategic level of enterprises [8]. Green innovation strategy has become the key way for enterprises to promote green transformation [9]. GIS includes enterprises' consideration of reducing the negative impact on the environment in the whole process of raw material procurement, raw material use and waste disposal [10]. The traditional logic holds that green innovation strategy requires special resource input, which will increase the cost of enterprises and is a kind of economic waste. However, the practice of some enterprises has proved that the green innovation strategy not only does not reduce its income but also improves corporate performance. Does green innovation strategy have to be at the expense of economic interests? The experiences of some enterprises show that profits can be improved through green marketing and sales of waste products and environmentally friendly technologies to other enterprises. Others can avoid environmental penalties, save on raw materials, and reduce waste disposal costs by improving their production processes [11]. In view of this, this study tries to achieve environmental protection and corporate income simultaneously through green innovation strategy.

Green innovation performance refers to enterprise's improvement of their product design or production process in terms of environmental protection and environmental management. Green innovation performance includes green product innovation performance and green process innovation performance from the perspective of innovation objects. The improvement of pollution prevention, energy saving, non-toxic or green product design, waste recycling, and so on, in product innovation and manufacturing process innovation enable enterprises to gain first-mover advantage and differentiated competitive advantage in the market [12]. There has also been previous literature about the content of green innovation performance, including enterprise economic performance, enterprise environmental performance, and enterprise social performance; this view focuses more on developing or adopting new technologies to add economic and social value to an enterprise [13]. Some scholars believe that the evaluation of green innovation performance should also consider the utilization of talents, equipment, and asset circulation, in addition to economic and environmental benefits, so as to reflect the favorable impact of saving on production cost and improving utilization efficiency [14]. There are also views that environmental innovation performance can be divided into indirect performance, direct performance, and knowledge output level [15]. Considering that the green innovation performance brought by strategic influence may not be reflected in the financial performance in the short term, but may be reflected in the green product, green process, green knowledge accumulation, and other aspects, this paper adopts the viewpoints of Banerjee, et al. (2003) [16], Leonidou et al. (2017) [17], Zameer et al. (2020) [18], and Panet al. (2017) [12], and measures GIP in two aspects: green product innovation and green process innovation.

Different from general organizational learning, organizational green learning focuses on learning and applying environmental protection knowledge based on green concept. As an internal factor, organizational green learning occurs under pressure or incentive [19]. The government's environmental regulation forces enterprises to adopt environment-friendly green technology, and the market's green demand stimulates enterprises to develop green products, resulting in enterprises having to organize employees to learn green knowledge, technology, and skills. It can be seen that organizational green learning lays more emphasis on green awareness and environmental protection knowledge learning. Organizational green learning is a crucial approach for enterprises to conduct a green innovation strategy. Through organizational green learning, enterprises master the current advanced green ideas and methods, and then apply them to enterprise green innovation. Through continuous learning, the original thinking, views, and cognition of enterprises can be updated, so as to change the original ideas and promote green innovation. Through green learning, enterprises can master cutting-edge theories and knowledge, help them make quick decisions, deal with changes outside, and improve their competitiveness [20].

Green technology refers to production equipment, methods, processes, product design and product delivery mechanisms that can save energy and natural resources, so as to reduce the environmental load of human activities [21]. Technological turbulence reflects the constant change in technology in an industry, which makes existing technology obsolete [22]. Green technology turbulence is an important external environmental factor, which describes the uncertainty and unpredictability of the market or industry, and represents the fuzziness and risk of green technology in the market [23]. Technology is a key point of innovation. In industries where technological turbulence is on the higher side, enterprises often encounter strong uncertainty about the expected results of green innovation [24].

Enterprises implementing green innovation strategy may allocate more resources in environmental management, making it easier for enterprises to develop green innovation strategy to obtain green innovation performance [25]. Previous research has highlighted the substantial benefits of green innovation strategy to enhance performance. Enterprises may decrease production costs and increase economic efficiency by applying environment-related innovation, such as reduction of energy consumption, reuse of material, and redefinition of the production process. Enterprises can further create corporate reputation and image to achieve green innovation performance [26]. Green innovation technology is complex and costly, and requires more environmental knowledge than traditional innovation. In order to implement green innovation strategy quickly, enterprises must constantly learn green knowledge. Organizational green learning requires enterprises to pay attention to the trend of creating and using green knowledge. Influenced by green innovation strategy, enterprises will consciously adopt green learning to promote green innovation behavior and improve green innovation performance [27]. Although researchers have studied the antecedents of green innovation strategy, there are few studies on the relationship between organizational green learning and green innovation performance, and the impact of organizational green learning on green innovation performance in the context of green technology turbulence still needs to be investigated.

In summary, the contributions of this study are as follows: First, based on the Natural Resource-Based View and taking "strategy-behavior-performance" as the logic, this study explores the impact of green innovation strategy on green innovation performance through green organizational learning, clarifies its path, and compensates for the deficiency of existing literature in explaining its internal mechanism. Secondly, based on the contingency theory, we discuss the boundary effect of green technology turbulence, which provides a theoretical and practical basis for enterprises to scientifically and effectively implement green innovation strategy in the context of green technological turbulence.

## 2. Theoretical Background and Hypotheses

### 2.1. Green Innovation Strategy and Green Innovation Performance

The theory of Natural Resource-Based View explains the green innovation behavior of enterprises related to environmental protection [28]. The natural environment can severely limit an enterprise's attempts to create sustainable advantages. An enterprise must be able to respond to changing environmental requirements by developing new resources; in other words, enterprises need nature environment-related resources and capabilities to build sustainable competitive advantage [29]. Pollution prevention, product management, and sustainable development are three important and interrelated strategic capacities [28]. As a strategy-driven orientation, green innovation strategy can guide enterprise behavior and achieve the goals of pollution prevention, product management, and sustainable development. Natural Resource-Based View theory emphasizes that by incorporating environmental considerations into strategy, enterprises can improve their ability to deal with the uncertainty of the connection between business operations and ecological issues, which is conducive to the development of competitive and scarce organizational capabilities [16,28,30].

Green innovation activities can enhance the profitability of enterprises by commercializing innovative products and processes [31]. Green innovation activities can not only minimize production pollution and increase productivity, but also gain competitive advantage by setting better prices for green products and improving corporate image [26]. Some scholars do not agree that green innovation strategy will inevitably lead to the improvement of enterprise performance; in particular, it is thought that it may damage the economic performance of enterprises in the short term. Enterprises' choice of green innovation strategy centered on the natural environment usually means higher initial cost and longer returns cycle [32]. However, in the long term, implementing green innovation strategies can help companies gain a reputation for environmentally based leadership and first-mover advantage. Considering that reputation itself is the source of market advantage, enterprises can take advantage of the first-mover advantage to occupy a long-term competitive advantage. Empirical research found that green innovation strategy can help enterprises win competitive advantage by gaining leadership reputation in environmental protection [33].

It is found that the green innovation strategy can lead to cost reduction, process improvement, and product innovation through a variety of green organization activities, and thus improve enterprise performance. Through strategic resource allocation, learning, and using new knowledge and ideas to creatively participate in green production, enterprises can improve the efficiency and effect of green product innovation or process innovation. Based on the above arguments, we propose that:

**Hypothesis 1 (H1).** *GIS is positively related to GIP.*

### 2.2. The Mediating Role of Organizational Green Learning

The Knowledge-Based theory holds that the degree of knowledge abundance among enterprises will directly determine the difference of competence and organizational learning, which help enterprises acquire knowledge resources [34]. Organizational learning is considered to be closely related to innovation performance [35]. Organizing green learning can improve enterprise environmental awareness and promote the smooth implementation of green innovation strategy [36]. Environmental knowledge learning can influence enterprise green innovation strategy by influencing enterprise decision [37]. Some studies have found that organizational green learning has a positive impact on enterprises' green innovation ability. Enterprises are facing a dramatically changing environment; only through the organization of green learning can they respond in a timely manner to the current changes to maintain the normal business environment of the enterprise [20]. Organizing green learning can transform green or clean ideas into business opportunities and improve the efficiency of existing products. Enterprises can also learn from the green

success of competitors and industry chain members. These continuous improvements require continuous learning and innovation of existing technologies and knowledge.

Organizational green learning is a key way for enterprises to conduct a green innovation strategy. Through organizational green learning, enterprises master the current advanced green ideas and methods, and then apply them to its green innovation practice. Green innovation strategy can promote environmental knowledge learning and stimulate green innovation behavior, so as to achieve green innovation performance [38]. That is, green innovation strategy can indirectly affect the green innovation performance of enterprises through organizational green learning. Accordingly, we propose that:

**Hypothesis 2 (H2).** *OGL mediates the relationship between GIS and GIP.*

**Hypothesis 2a (H2a).** *GIS is positively related to OGL.*

**Hypothesis 2b (H2b).** *OGL is positively related to GIP.*

*2.3. The Moderating Role of Green Technology Turbulence*

Contingency theory emphasizes that no one theory or method can be applied to all situations, and the "fit" between enterprise structure and specific uncertain environmental characteristics determines the performance of enterprises [39]. That is to say, there is a significant relationship between external environment and enterprise behavior and performance results. A turbulent technological environment has brought some difficulties and challenges to enterprise green innovation. Green technology turbulence has magnified the potential for unexpected risks. In the existing studies, green technology turbulence is an important situational condition for green innovative enterprises facing the uncertainty of the external environment, and plays an important role in strategy formulation and implementation. Therefore, this study regards green technology turbulence as a key boundary condition and explores its moderating role in the path relationship between green innovation strategy and green innovation performance. Thus, we hypothesize as follows:

**Hypothesis 3 (H3).** *GTT moderates the relationship between GIS and GIP.*

When considering green process innovation or green product innovation, especially when considering the transformation, redesign and creation of new products, technological change is the most relevant element in the innovation process [40]. Although higher turbulence of green technology will make existing technology and knowledge obsolete more quickly and weaken the competitive advantage, green organizational learning can obtain more new green knowledge and technology-related green innovation, make them more diverse, and strengthen the green differentiation advantage and green innovation performance of the enterprise. Therefore, we assume the following:

**Hypothesis 4 (H4).** *GTT moderates the relationship between OGL and GIP.*

We do not consider the moderating effect of green technology turbulence between GIS and OGL, because this study focuses more on the moderating effect of green technology turbulence on green innovation performance, which directly affects the implementation effect of green innovation strategy, and the moderating effect on organizational green learning is second in importance. Therefore, this study focuses on the second half of the model.

*2.4. Hypothesized Conceptual Model*

Based on the above discussion, although the academic research on green innovation strategy keeps increasing, the existing research has not paid much attention to the conditions and methods under which green innovation strategy can obtain green innovation

performance, and organizational green learning provides a way to solve this problem. Both the implementation of green innovation strategy and the realization of green innovation performance are quite complex tasks, which usually require information and skills different from traditional industry knowledge. In inter-organizational practice, organizational green learning can help enterprises acquire, disseminate, interpret, and use information from suppliers or customers, and understand, master, and transform knowledge conducive to green innovation in a timely manner [35]. Green technology turbulence is an important situational condition for enterprises facing the uncertainty of the external environment. Therefore, this study introduces organizational green learning as a mediator variable and green technology turbulence as a moderator variable to explore the path mechanism between green innovation strategy and green innovation performance. The hypothesized conceptual model proposed in this study is shown in Figure 1. The hypotheses were proposed as follows:

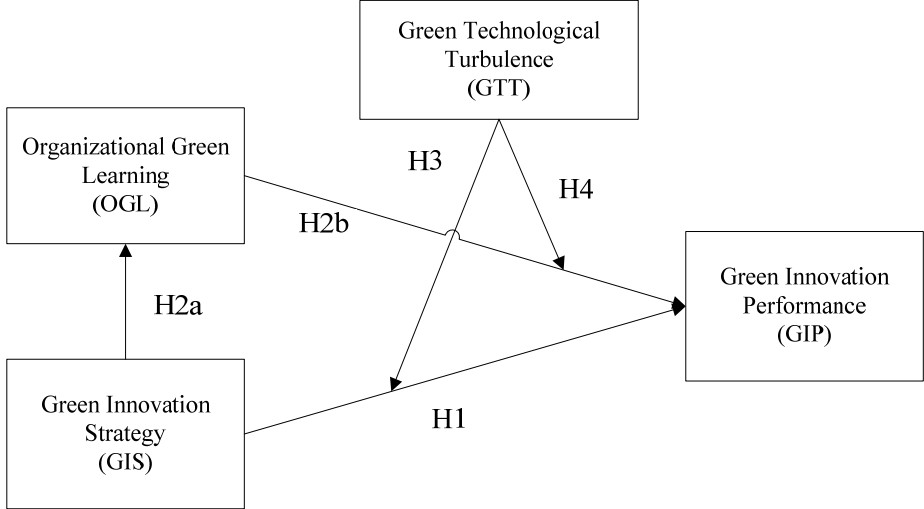

**Figure 1.** Hypothesized conceptual model.

## 3. Methodology

### 3.1. Data Source and Sample

This study collected data by issuing research questionnaires to manufacturing enterprises that were willing to accept the survey [41].The manufacturing industry shows a strong environmental sensitivity in enterprise environmental strategy and practice, and manufacturing enterprises have a great influence and dependence on natural resources and material resources [42], and are often regarded as one of the biggest "culprits" in­causing environmental deterioration. Manufacturing enterprises also have the ability to minimize negative impacts, and even become a major contributor to solving environmental problems [43].The measurement items in this paper are mainly obtained through the respondents' subjective feelings. Therefore, before the formal investigation, we tried to adopt various measures and methods to improve the quality of the questionnaire measurement, so as to ensure the scientific nature and objectivity of the sample data.For measuring the core variables involved, maturity scales were used for reference or adaptation, and a 7-point Likert scale was used for evaluation.We modified the foreign scales by translating them back from Chinese to English until the two transformations showed no substantial difference in the meanings of the scales.The project team used expert consultation and a pre-survey of 80 MBA students to refine the initial questionnaire. Finally, we contacted 578 manufacturing enterprises, 318 of whichwere willing to answer the questionnaire, and two of whichwere incomplete. Valid data were 316, accounting for 54.7%.

### 3.2. Measurement of Variables

The scale used in the study was designed for the 4 variables. The 4 variables that needed to be operationalized were: green innovation strategy, organizational green learning, green technology turbulence, and green innovation performance. The latent variable measurement scales were all from the mature scale (Appendix A), and Chinese-English loop twice translating method was adopted to ensure the reliability and validity of our scale. Likert seven-point scoring method was selected for measurement, in which "1" meant "completely inconsistent" and "7" meant "completely consistent". Respondents were required to evaluate and select the corresponding questions according to their actual situation of enterprise [44].

For the green innovation strategy, we adopted the viewpoint of Chen (2006) and included 7 questions. A sample item is: "we adjusted business activities to reduce the damage to the ecological environment" [27].

For organizational green learning, we adopted the 7-item organizational green learning scale developed by Dai Wanliang (2020) [20] for reference to the research results of Atuahene et al. (2007) [45], among which 4 items measured the exploitative organizational green learning; a sample item is: "we will pay attention to more environmentally friendly production processes when developing new products". A total of 3 items measured exploratory organizational green learning; a sample item is: "One of the reasons we seek information is to learn more about environmental protection".

For green technology turbulence, we used Sheng et al. (2011) [46] for reference and Jiang et al. (2018) [47] and Wei et al. (2020) [23] for comprehensive reference to measure the turbulence of green technology. A total of 4 items were used, and a sample item is: "The green technology in our industry is changing rapidly".

Green innovation performance in this study referred to two dimensions, green product innovation performance and green process innovation performance. Relatively speaking, the effect was relatively intuitive, and most managers were well aware of this part of the enterprise. We referenced Banerjee et al. (2003) [16] and Leonidou et al. (2017) [17] to measure green innovation performance in terms of green product innovation and green process innovation with a total of 10 questions, and made appropriate corrections and adjustments based on Zameer et al. (2020) [18]. A sample item is: "We have developed new products or services for environmental management in the past two years".

### 3.3. Control Variables

The consideration is that enterprises of different sizes may have different capabilities to guarantee the implementation of a green innovation strategy, while managers' attitudes to green innovation strategies may vary with the age of enterprises. We took firm size and firm age as control variables to control the impact of green innovation strategy on green innovation performance. Firm size was measured by the number of employees. Firm age was the years since the establishment of the enterprise [48].

### 3.4. Reliability and Validity

In the reliability test, Cronbach's $\alpha$ value and combined reliability (CR) were used to a make comprehensive judgment. As shown in Table 1, the test results of the data show that the $\alpha$ value of each latent variable is between 0.892–0.974. The CR value is between 0.893–0.974, the above index values are all better than 0.8, indicating that the internal consistency between latent variable measurement questions is very good, and the measurement reliability is ideal. In all terms of validity test, the measurement tools of core concepts in this study were adapted from mature research scales, so that the content validity of the measurement could be guaranteed. As shown in Table 1, the results of the confirmatory factor analysis show that the factor loads of the core concept measurement questions in the study are all higher than 0.6, and the average extraction variance (AVE value) is significantly higher than 0.6, indicating that the convergence validity level of core concept measurement is also ideal.

**Table 1.** Reliability and validity indicators.

| Variables | Items | Loading | Cronbach's $\alpha$ | CR | AVE |
|---|---|---|---|---|---|
| GIS | GIS1 | 0.859 | 0.964 | 0.964 | 0.793 |
| | GIS2 | 0.883 | | | |
| | GIS3 | 0.885 | | | |
| | GIS4 | 0.903 | | | |
| | GIS5 | 0.901 | | | |
| | GIS6 | 0.902 | | | |
| | GIS7 | 0.901 | | | |
| OGL | OGL1 | 0.881 | 0.965 | 0.966 | 0.801 |
| | OGL2 | 0.914 | | | |
| | OGL3 | 0.898 | | | |
| | OGL4 | 0.847 | | | |
| | OGL5 | 0.909 | | | |
| | OGL6 | 0.905 | | | |
| | OGL7 | 0.908 | | | |
| GTT | GTT1 | 0.876 | 0.892 | 0.893 | 0.681 |
| | GTT2 | 0.605 | | | |
| | GTT3 | 0.867 | | | |
| | GTT4 | 0.915 | | | |
| GIP | GIP1 | 0.893 | 0.974 | 0.974 | 0.791 |
| | GIP2 | 0.919 | | | |
| | GIP3 | 0.904 | | | |
| | GIP4 | 0.915 | | | |
| | GIP5 | 0.905 | | | |
| | GIP6 | 0.895 | | | |
| | GIP7 | 0.864 | | | |
| | GIP8 | 0.899 | | | |
| | GIP9 | 0.851 | | | |
| | GIP10 | 0.843 | | | |

Note: Fit statistics: $\chi2$ = 935.277, $p$ = 0.00, df = 344, $\chi$/df 2 = 2.719, NFI = 0.92, RFI = 0.912, CFI = 0.948, RMSEA = 0.074.

*3.5. Common Method Variance*

Collecting data in the form of questionnaires will inevitably lead to the problem of common method bias. According to the suggestions of Podsakoff et al. (2003) [49], we adopted two methods of pre-control and post-test to reduce the influence of common method bias. In terms of prior control, the questionnaire design emphasized the methods such as no right or wrong answers, anonymity, and academic only, so as to reduce common method bias and social desirability bias. In terms of post-mortem test, Harman single factor test was first used to evaluate the factor structure of variables. Exploratory factor analysis (EFA) showed that after adding all the items of constructs into the principal component analysis, the unrotated factors solutions showed three separate factors with eigenvalues above 1.0, which explained 80% of the total variance, and KMO was 0.975. Secondly, a confirmatory factor analysis (CFA) was performed to examine the possible impact of common method bias, in which all indicators in the initial measurement validation were limited to a single factor. The fitting index of the model was poor: $\chi2$/ DF =2.719, RMSEA = 0.074, CFI = 0.948, TLI = 0.943, and IFI = 0.948. Therefore, there is no serious problem of common method bias in this study.

## 4. Results

*4.1. Characteristics of Samples*

In the process of data collection, we did not limit the hierarchy of managers, because this study mainly examined the implementation process and results of green innovation strategy, and middle managers and junior management is more direct. Therefore, the data mainly came from middle and junior managers. The types of enterprises included high

pollution manufacturing and low pollution manufacturing. The specific conditions of the sampled enterprises in this study are presented in Table 2.

**Table 2.** Characteristics of samples.

| Characteristics | Categories | Frequency | Percentage (%) |
|---|---|---|---|
| Position | Top manager | 24 | 7.59 |
| | Middle manager | 73 | 23.1 |
| | Junior managers | 219 | 69.3 |
| Listed enterprise | Yes | 107 | 33.86 |
| | No | 209 | 66.14 |
| Firm age | 1–3 years | 49 | 15.51 |
| | 4–10 years | 79 | 25 |
| | 11–20 years | 99 | 31.33 |
| | 21–30 years | 39 | 12.34 |
| | More than 31 | 50 | 15.82 |
| Number of employees | Under 100 | 108 | 34.18 |
| | 100–500 | 69 | 21.84 |
| | 501–1000 | 36 | 11.39 |
| | More than 1000 | 103 | 32.59 |
| Ownership structure | Private firms | 157 | 49.68 |
| | Collective and State-owned firms | 122 | 38.61 |
| | Foreign-funded firms | 37 | 11.71 |
| Industry | High pollution industry | 228 | 72 |
| | Low pollution industry | 88 | 28 |

*4.2. Hypothesis Testing*

We used Pearson correlation coefficient analysis to examine the correlation strength and direction of the relationship among variables from the size of the correlation coefficient [50]. The results are shown in Table 3. Except that the correlation coefficients between control variables and some variables are not significant, the correlation coefficients between any two variables or dimensions are significant; this indicates that the hypothesis proposed in this study is reasonable to a certain extent and can be further tested.

**Table 3.** The descriptive analysis and correlation coefficients.

| | M | SD | 1 | 2 | 3 | 4 | 5 | 6 |
|---|---|---|---|---|---|---|---|---|
| 1 GIS | 5.446 | 1.555 | 1 | | | | | |
| 2 OGL | 5.345 | 1.513 | 0.732 ** | 1 | | | | |
| 3 GTT | 5.114 | 1.480 | 0.696 ** | 0.773 ** | 1 | | | |
| 4 GIP | 5.308 | 1.498 | 0.775 ** | 0.833 ** | 0.871 ** | 1 | | |
| 5 Firm age | 2.880 | 1.272 | 0.139 * | 0.071 | 0.127 * | 0.129 * | 1 | |
| 6 Firm size | 2.420 | 1.259 | 0.105 | 0.059 | 0.126 * | 0.098 | 0.490 ** | 1 |

Note: Firm age and firm size are control variables. ** $p < 0.01$, * $p < 0.05$.

In this study, Spss26.0, Amos25.0, and Process plug-in software were used to test the hypothesis using hierarchical regression and Bootstrap method.

4.2.1. The Mediating Effect of OGL

We adopted bootstrap of Spss to test the mediating effect of OGL on the relationship between GIS and GIP. Table 4 shows the test results of the main effect and mediating effect. The results show that direct effect of GIS on GIP is significant ($t$ = 21.371, $p < 0.01$), and the

direct effect of GIS on GIP is still significant ($t$ = 8.381, $p$ < 0.01) after the mediation variable OGL is added. Therefore, H1 is supported.

**Table 4.** Regression results of the mediating effect of OGL.

| | GIP | | GIP | | OGL | |
|---|---|---|---|---|---|---|
| | $t$ | $p$ | $t$ | $p$ | $t$ | $p$ |
| OGL | 13.958 | 0.000 | | | | |
| GIS | 8.381 | 0.000 | 21.371 | 0.000 | 18.901 | 0.000 |
| Firm age | 1.065 | 0.288 | 0.421 | 0.674 | −0.671 | 0.503 |
| Firm size | 0.335 | 0.738 | 0.213 | 0.832 | −0.081 | 0.935 |
| R-sq | 0.755 | | 0.602 | | 0.537 | |
| F | 239.607 | | 157.004 | | 120.387 | |

In Table 5, there is no 0 between boot LLCI and boot ULCI (95%CI) in the direct effect of GIS on GIP and the mediation effect of OGL, indicating that GIS can not only directly affect GIP, but also affect GIP through the mediation effect of OGL. The direct effect (0.336) and mediating effect (0.408) accounted for 54.9 and 45.1% of the total effect (0.744), respectively. Therefore, H2a and H2b are supported.

**Table 5.** Total effect, direct effect, and intermediate effect.

| | **Coeff** | **Boot SE** | **Boot LLCI** | **Boot ULCI** | **Percent** |
|---|---|---|---|---|---|
| Intermediate effect | 0.408 | 0.043 | 0.658 | 0.827 | 54.9% |
| Direct effect | 0.336 | 0.071 | 0.203 | 0.479 | 45.1% |
| Total effect | 0.744 | 0.063 | 0.290 | 0.536 | |

Note: Boot LLCI and Boot ULCI are under 95%CI.

### 4.2.2. The Moderating Effect of GTT

Mediated mediation analysis can be variously described as conditional process analysis, conditional process model, or the hybrid model [51]. We adopted model 15 in Process, which was consistent with the theoretical model of this study, assuming that the latter half of the mediation model and its direct path were adjusted, to test the moderate effect of GTT on the direct effect and mediating effect of GIS to GIP under the condition of controlling enterprise age and enterprise size.

The results of Table 6 show that after GTT is put into the model, the product term of OGL and GTT have a significant effect on the relationship between OGL and GIP ($t$ = −0.042, $p$ < 0. 05), while it is not significant on the direct effect of GIS and GIP ($t$ = 0.023, $p$ > 0.05). This indicates that GTT has a significant moderating effect on the mediation relationship between OGL and GIP, but shows no significant moderating effect on the direct effect between GIS and GIP. Thus, these results show that H4 is supported and H3 is not supported.

We further examined the details of the significant moderating effects following Aiken and West's (1991) suggestions [52,53]. In Table 7 and Figure 2, simple slope analysis shows that OGL has a significant positive effect on GIP when GTT level is high (M + 1SD), simple slope = 0.214, $t$ = 0.053, $p$ < 0.001. When GTT level is low (M − 1SD), OGL also has a positive effect on GIP, but its effect is small, simple Slope = 0.339, $t$ = 0.046, $p$ < 0.001, indicating that with the increase of GTT level, the effect of OGL on GIP gradually increases.

**Table 6.** Conditional process analysis.

| | GIP | | |
| | coeff | se | *t* |
|---|---|---|---|
| constant | 5.303 ** | 0.093 | 57.326 ** |
| GIS | 0.212 ** | 0.036 | 5.931 ** |
| OGL | 0.276 ** | 0.039 | 7.009 ** |
| GTT | 0.492 ** | 0.038 | 12.95 ** |
| GIS × GTT | 0.023 | 0.02 | 1.179 |
| OGL × GTT | −0.042 ** | 0.02 | −2.126 ** |
| Firm Size | −0.026 | 0.031 | −0.835 |
| Firm Age | 0.036 | 0.031 | 1.172 |
| R-sq | 0.844 | | |
| F | 336.328 | | |

Note: ** $p < 0.01$, * $p < 0.05$.

**Table 7.** Mediating effects at different GTT levels.

| | Effect | *t* | Boot SE | Boot LLCI | Boot ULCI |
|---|---|---|---|---|---|
| M − 1SD | 0.339 ** | 0.046 ** | 7.433 | 0.249 | 0.429 |
| M | 0.276 ** | 0.039 ** | 7.009 | 0.199 | 0.354 |
| M + 1SD | 0.214 ** | 0.053 ** | 4.058 | 0.11 | 0.317 |

Note: Boot LLCIand Boot ULCI are under 95%CI. ** $p < 0.01$, * $p < 0.05$.

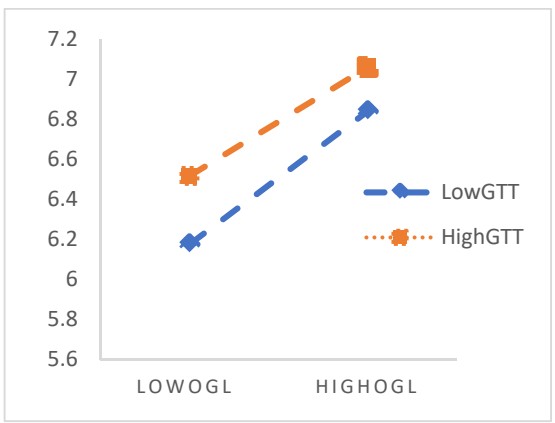

**Figure 2.** Interactive effects of GIS and GTT on GIP.

Based on the above data, the results of hypothesis testing are shown in Table 8.

**Table 8.** Results of hypotheses tests.

| Hypotheses | Outcome |
|---|---|
| H1: GIS is positively related to GIP. | Supported |
| H2: OGL mediates the relationship between GIS and GIP. | Supported |
| H2a: GIS is positively related to OGL. | Supported |
| H2b: OGL is positively related to GIP. | Supported |
| H3: GTT moderates the relationship between GIS and GIP. | Not supported |
| H4: GTT moderates the relationship between OGL and GIP. | Supported |

## 5. Discussion

Our empirical analysis shows that green innovation strategy plays a significantly supportive role in promoting green innovation performance; this finding is consistent with the studies by Sun et al. (2021) [50,53,54]. Additionally, we find that organization of green learning partly mediates the relationship between green innovation strategy and green innovation performance. On the one hand, the green innovation strategy requires

enterprises to study green related fields and try to acquire new green knowledge and information. On the other hand, the patents and skills obtained by the organization through green learning can further improve the green performance of the organization; they directly affect the effectiveness of green product innovation and green process innovation of employees, and then influence an increase in the number of green products and the efficiency of green processes [55]. Enterprises with a strong organizational green learning ability are able to keep up with trends in the outside world and use the latest technology to meet new green needs and shape green competitiveness. Conversely, enterprises with weak organizational green learning skills will miss this opportunity. This provides a theoretical explanation for why enterprises adopting similar green innovation strategies show different business results. This is also consistent with the research of Tu and Wu (2021) [56].

Green technology turbulence reflects the challenge and impact of the uncertainty of external green technology changes on the existing green innovation activities of enterprises [57]. It represents the innovation speed and frequency of key green technologies in an industry. In this study, green technology turbulence plays a positive moderating role in the mediating relationship between organizational green learning and green innovation performance. Based on contingency theory, this moderated mediation model reveals how organizational green innovation strategy affects green innovation performance under the condition of green technology turbulence. The more intense the turbulence of green technology, the greater the promotion effect of organizational green learning on green innovation performance, just as Ogbeibu et al. (2020) [58] previously mentioned. When the industry technology environment is in a state of high-speed turbulence, the rapid change of technology will shorten the life cycle of existing products, quickly eliminate the current dominant products or services, and weaken the existing competitive advantage. Currently, the rapid update of industry technology also forces enterprises to break through technical difficulties quickly and invent new technologies with a higher success rate. If an enterprise can organize green learning to obtain more green-related new technologies, it will produce more green product, just as was previously proposed by Wang (2020) [57], Thornhill (2006) [59] and Yang (2018) [60]. Some scholars believe that in a relatively stable technological environment, enterprises tend to seek more profits from existing technologies and markets in order to reduce risks, so the innovative technological achievements of enterprises will also be reduced. This is consistent with the results of Yin (2014) [61].

However, the moderating effect of GTT on the direct effect between GIS and GIP is not supported. To gain insight into this counterintuitive result, we conducted further interviews with managers at some manufacturing enterprises. As for the non-significant moderating effect of GTT, respondents explained that when GTT is low, the external technical environment is relatively stable and the execution effect of the green innovation strategy is stable, and there is a steady consumer demand for green products in the market. In this case, enterprises can rationally allocate resources according to established strategies, conduct green development, implement green production, conduct green marketing, breed green corporate culture, and then achieve green performance. Although in the case of high GTT level, it is more difficult for enterprises to gain new green technology support, enterprises that have implemented green innovation strategy still achieve better performance based on existing green advantages than those that have not implemented green innovation strategy. Therefore, the moderating effect of green technology turbulence between green innovation strategy and green innovation performance is not significant. However, the impact of green innovation strategy on green innovation performance through organizational green learning can be significantly moderated by green technology turbulence. Another possible explanation is that despite the importance of GTT, enterprises implementing green innovation strategies do not care much about external green technology turbulence due to their enthusiasm for pursuing green market premiums. Several scholars have also suggested that enterprises must organize their product development teams to be more agile and responsive to technological change [62]. Changing technologies may limit the performance of breakthrough innovations and fail to incorporate new technologies into

products and provide new benefits to customers [63,64]. In the case of rapid technological environment turbulence, the weak technological advantage brought by some enterprises' green strategy can easily be offset by external technological changes. Therefore, it is possible that the moderating effect of green technology turbulence is different under different environmental backgrounds.

## 6. Conclusions and Implications

### 6.1. Conclusions

The pressure of sustainable development and the ecological environment keep increasing, which urges enterprises to continuously practice green innovation strategy, improve their social reputation as responsible enterprises, and gain first-mover advantage. This study explains the path from green innovation strategy to green innovation performance through empirical evidence, and illustrates the moderating role of green technology turbulence, and thus supplements the research of green innovation strategy. We studied the impact of green innovation strategies on 316 selected manufacturing firms and reported significant results. The first conclusion is that the practice of green innovation strategy can bring green innovation performance for enterprises. Both the government and the market should encourage enterprises to formulate and implement green strategies through appropriate and positive environmental policies, such as green subsidies, green financing, green product premium, and so on. Secondly, organizational green innovation plays a mediating role in the process of obtaining performance of green innovation strategy.

The second conclusion is that organizational green learning plays a partially mediating role in the relationship between green innovation strategy and green innovation performance. Managers should concentrate on the latest green-related knowledge and information inside and outside the organization, and even the green success of competitors [65], in order to improve the performance of incremental innovation in times of technological turbulence. Enterprises can also establish platforms (such as inter-enterprise social media) for external and internal stakeholders to communicate and cooperate with each other by sharing green knowledge and information. In this way, enterprises can gain more green advantages.

The third conclusion is that green technology turbulence moderates the relationship between organizational green learning and green innovation performance. To the best of our knowledge, this study is the first attempt to incorporate organizational green learning and green technology turbulence into the study, forming a moderated mediation model, which helps to clarify the mixed results of the impact of green innovation strategy on innovation performance in previous studies. In the case of rapid turbulence of green technology, enterprises with green innovation strategy show better learning ability in green product innovation and green process innovation, so as to form positive green competitiveness [57].

### 6.2. Implications

The findings of this study have implications for practitioners as well as some policy implications. Our empirical results show that green innovation strategy has a positive influence on green innovation performance. From the practical perspective, this research model provides a more comprehensive understanding for managers of enterprises who want to enhance green innovation performance. Managers who want to achieve better results from their sustainability initiatives should adopt a green innovation strategy to achieve efficient resource usage, $CO_2$ emissions reduction, energy utilization reduction, moderate cost incurred, ethical waste management, and natural resources conservation. This result is consistent with a survey on green innovation strategy in Indonesia, which has shown that managers need to start with developing a green innovation strategy to improve green performance [66]. Moreover, this study also clearly shows managers that having a strategy is not sufficient to directly enhance green innovation performance. Managers need to find ways to effectively implement green innovation strategies, encourage employees

to conduct organizational green learning, and remain in touch with advanced green technology capabilities, so as to maintain the enterprise in a leading position in the turbulent technological environment.

Our findings may be of value to inform policy maker's efforts in promoting sustainable development goals. The results of this research contribute to providing an approach on how to conduct a better environmental management, which brings more benefits for a better life in society and the world. Manufacturing enterprises are the major contributors to environmental damage. The enterprises with the green innovation strategy Proactive reduce emissions and start conducting resource efficiency. When most manufacturing enterprises implement this model to solve environmental issues, the whole of society will gain more benefits from the reduction of environmental degradation, the availability of more green products, the improvements in resource efficiencies and economic development, and the betterment of life quality. Policy makers should guide and encourage enterprises to adopt green innovation strategies, and help enterprises with green operation through green subsidies, green financing, and exclusive channels for green products.

*6.3. Limitations and Future Research*

Despite the rigor of this study, some limitations need to be considered when interpreting the results and conducting future research. First, this study emphasizes that manufacturing enterprises are more involved in environmental practices. Manufacturing enterprises consume more resources, and the performance brought by green innovation is more obvious than that of other industries. However, enterprises in other industries are also facing the decision of green innovation strategy. Future research may explore the effect of green innovation strategy on green innovation performance in different industries to enrich the research on green strategy research. In addition, due to the consideration of the availability of data and the workload of this study, this study focuses on the moderating effect of green technology turbulence on the latter half of the model. For further research, our research team will consider the moderating effect of green technology turbulence on the relationship between green innovation strategy and organizational green learning. Finally, in order to investigate the actual situation of enterprises accurately and effectively, our research obtained primary data through questionnaire survey, which may be affected by subjective perception of subjects despite our efforts to eliminate this effect. Therefore, future studies may explore the quantitative measurement of variables to reconfirm the research conclusion of this paper.

**Author Contributions:** Conceptualization, N.W. and J.Z.; data curation, W.W.; formal analysis, N.W.; funding acquisition, X.Z.; investigation: N.W., J.Z., X.Z. and W.W.; methodology, N.W., J.Z. and X.Z.; project administration, W.W. and N.W.; writing—original draft: N.W., J.Z., X.Z. and W.W.; writing—review and editing: N.W., J.Z., X.Z. and W.W. All authors have read and agreed to the published version of the manuscript.

**Funding:** This research was funded by The National Social Science Foundation of China (20BGL059).

**Informed Consent Statement:** Informed consent was obtained from all subjects involved in the study.

**Data Availability Statement:** Some or all data and models that support the findings of this study are available from the corresponding author upon reasonable request.

**Conflicts of Interest:** The authors declare no conflict of interest.

## Appendix A

**Table A1.** Measurement scales.

| Constructs | Item Description | Source |
|---|---|---|
| GIS | GIS1 We adjusted business activities to reduce the damage to the ecological environment<br>GIS2 Although government regulations did not require it, we voluntarily took environmental remediation actions<br>GIS3 We adjusted our operations to reduce waste of resources and emissions of pollutants<br>GIS4 We adjusted our operations to achieve recycling of non-renewable raw materials, chemicals, and components<br>GIS5 We replaced traditional fuels with some new and less polluting sources of energy<br>GIS6 We adjusted our operations to reduce energy consumption<br>GIS7 We adjusted our operations to reduce the environmental impact of our products | Chen (2006) |
| OGL | OGL1 One of the purposes of our search for information is to find more energy-efficient solutions to problems<br>OGL2 One of the purposes of our search for information is to ensure that we save energy and reduce emissions and environmental pollution<br>OGL3 We pay attention to more environmentally friendly production processes when developing new products<br>OGL4 We tend to use environmental knowledge that is relevant to existing projects<br>OGL5 One of the purposes of our search for information is to learn more about environmental protection<br>OGL6 One of the purposes of our search for information is to develop new green projects and enter new markets<br>OGL7 We collect information that is greener than technology experience in existing markets | Dai Wanliang, et al. (2020); Atuahene, et al. (2007) |
| GTT | GTT1 Green technology in our industry is changing fast<br>GTT2 The direction of green technology in our industry is hard to predict<br>GTT3 Most green technology innovations in our industry are radical changes to existing technologies<br>GTT4 The green technology revolution in our industry has created many opportunities | Sheng, et al. (2011); Jiang, et al. (2018); Wei, et al. (2020) |
| GIP | GIP1 We have developed new products or services in environmental management in the past two years<br>GIP2 We have selected less polluting product materials for product development or design<br>GIP3 We have selected the product materials that consume the least energy and resources for product development or design<br>GIP4 We have used minimal materials to compose products for product development or design<br>GIP5 In the process of product development or design, we have carefully considered whether the product is easy to recycle, reuse, and decompose<br>GIP6 We have adopted new methods of environmental management after conventional methods failed<br>GIP7 The production process of enterprises has effectively reduced the discharge of harmful substances or waste<br>GIP8 We have recycled waste and discharge in the production process so that these can be treated and used<br>GIP9 Our production processes consume less water, electricity, coal, or oil<br>GIP10 Our production process reduces the use of raw materials | Banerjee, et al. (2003); Leonidou et al. (2017); Zameer et al. (2020) |

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
