# Peer review of "How to Improve Green Innovation Performance: A Conditional Process Analysis"

_sustainability, doi:10.3390/su14052938_

Round 1
Reviewer 1 Report
The paper deals with ways of improving green innovation performance, and more specifically with a green process analysis, as enterprises are faced with environmental issues posing the challenge of shifting to green enterprises, and delivering green innovation performance, and green product innovation.
The summary delivers a good framework by presenting from the onset the approach, and intentions of the authors, and the pursued goals of the paper, while it also touches on the findings and conclusions of the paper.
The introduction is comprehensive and details well the arguments for the selected approach. The theoretical background is sound, but the authors should have considered a more clearly delineated specialized literature review for substantiating their approach. Even though there are rich specialized literature references, it would have been a better option to have at least some better structured references, that would also assist in understanding how the hypotheses were built.
The methodology was chosen according to the objectives and the collected data, and has the merit of relying on comprehensively processed primary sources. The use and interpretation of questionnaires that support the construction of the analysis model is very well explained and minutely realized. The use of primary data for testing the hypotheses and delivering on the stated purpose of the paper is a valuable contribution and useful for understanding the process inside manufacturing companies.
The step-by-step approach elaborates generously on how the analysis framework was constructed, and how the results were interpreted as to avoid common method bias, based on two methods of pre-control and post-test. However, a suggestion would be to separate clearly the results and their interpretation from the discussion, to provide the reader a clearer understanding of the sections.
The analysis is explained comprehensively, and the data in the annex help in understanding better the approach chosen by the authors.
The conclusions seem a bit too short and it would have been recommendable for the authors to develop more on the main findings reflected in the conclusions, that should be a little bit restructured. The paper would benefit from such improvement, as the insights could deliver more on the usefulness of the approach also for enterprises from other industries than manufacturing.
The paper is recommended for publishing, but the authors should operate some minor revisions, as to integrate better the parts concerned with methodology, methods, and results obtained based on the partial least squares and regression analysis.
The paper is written in a clear, comprehensive and accessible manner. Nonetheless, some minor language revisions are recommended.
Author Response
Response to Reviewer 1 Comments
Reviewer 1
Title: How to Improve Green Innovation Performance: a Conditional Process Analysis (Sustainability-1618595)
The paper deals with ways of improving green innovation performance, and more specifically with a green process analysis, as enterprises are faced with environmental issues posing the challenge of shifting to green enterprises, and delivering green innovation performance, and green product innovation.The summary delivers a good framework by presenting from the onset the approach, and intentions of the authors, and the pursued goals of the paper, while it also touches on the findings and conclusions of the paper.
Response: We very much appreciate your constructive comments and thank you for giving us an opportunity to revise and resubmit our study. We believe that your guidance and comments have indeed assisted us significantly in improving our research and its potential contribution. Following your suggestions, we made a number of changes throughout the manuscript, increased precision in each section and clarified the issues you raised. We believe that we have made our best efforts and provided the optimal solution to your concern and justification for our approach. We hope that you will find our ‘response points’ thorough and convincing. In this letter, we include your original comments in black and then set forth how we have dealt with your comments in red in the new, revised manuscript.
Point 1: The introduction is comprehensive and details well the arguments for the selected approach. The theoretical background is sound, but the authors should have considered a more clearly delineated specialized literature review for substantiating their approach. Even though there are rich specialized literature references, it would have been a better option to have at least some better structured references, that would also assist in understanding how the hypotheses were built.
Response 1: Thank you for your suggestion. We agree with your point and we have revised the paper in the Introduction, to enrich the structured literature review about the approach for building the hypotheses. We added this in the last paragraph of the introduction. The specific modifications are as follows:
Enterprises implementing green innovation strategy may allocate more resources in environmental management, making it easier for enterprises to develop green innovation strategy to obtain green innovation performance [22]. Previous research has highlighted the substantial benefits of green innovation strategy to enhance the performance. Enterprises may decrease production costs and increase economic efficiency by applying environment-related innovation, such as reduction of energy consumption, reuse of material and redefinition of production process. Enterprises can further create corporate reputation and image to achieve green innovation performance [23]. Green innovation technology is complex and costly and requires more environmental knowledge than traditional innovation. In order to implement green innovation strategy quickly, enterprises must constantly learn green knowledge. Organizational green learning requires enterprises to pay attention to the trend of creating and using green knowledge. Influenced by green innovation strategy, enterprises will consciously adopt green learning to promote green innovation behavior and improve green innovation performance [24]. Although researchers have studied the antecedents of green innovation strategy, there are few studies on the relationship between organizational green learning and green innovation performance, and the impact of organizational green learning on green innovation performance in the context of green technology turbulence still needs to be investigated.
- Huang J; Li Y. How resource alignment moderates the relationship between environmental innovation strategy and green innovation performance. The Journal of business & industrial marketing. 2018, 33, 316-324. https://doi.org/10.1108/JBIM-10-2016-0253
- Chen, Y S.; Lai, S. B. et al. The Influence of Green Innovation Performance on Corporate Advantage in Taiwan. Journal of Business Ethics, 2006, 67(4), 331-339. https://doi.org/10.1007/s10551-006-9025-5
- Wang J, Xue Y, Sun X, Yang J. Green learning orientation, green knowledge acquisition and ambidextrous green innovation. Journal of cleaner production. 2020,250,119475. https://doi.org/10.1016/j.jclepro.2019.119475
Point 2: The step-by-step approach elaborates generously on how the analysis framework was constructed, and how the results were interpreted as to avoid common method bias, based on two methods of pre-control and post-test. However, a suggestion would be to separate clearly the results and their interpretation from the discussion, to provide the reader a clearer understanding of the sections.
Response 2: Thank you for making this comment. We have revised the paper in section ‘4.Results and Discussion’ on page 8 and 9, we have taken two actions to address your comment.
First, we removed the word ‘discussion’ from heading ‘4.Results and Discussion’, considering the heading ’ 5.Discussion.’. Here again, we very much appreciate your detailed review.
Second, we have further separated the results and their interpretation from the discussion clearly.
Point 3: The conclusions seem a bit too short and it would have been recommendable for the authors to develop more on the main findings reflected in the conclusions, that should be a little bit restructured. The paper would benefit from such improvement, as the insights could deliver more on the usefulness of the approach also for enterprises from other industries than manufacturing.
Response 3: Thank you for your suggestion. We agree with your point and have revised the paper in the part of ‘6.conclusions and implications’. We added the section ‘6.2. Implications’ and the specific modifications are as follows:
6.2. Implications
The findings of this study have implications for practitioners as well as some policy implications. Our empirical results show that green innovation strategy has positive influence on green innovation performance. From the practical perspective, this research model provides a more comprehensive understanding for managers of enterprises who want to enhance green innovation performance. Managers who want to get better results from their sustainability initiatives should adopt green innovation strategy to achieve efficient resource usage, CO2 emissions reduction, energy utilization reduction, moderate cost incurred, ethical waste management and natural resources conservation. This result is consistent with a survey on green innovation strategy in Indonesia which has shown that managers need to start with developing a green innovation strategy to improve green performance [68]. Moreover, this study also clearly shows managers that having a strategy is not sufficient enough to directly enhance green innovation performance. Managers need to seek approaches on how to effectively implement green innovation strategy, encourage employees to carry out organizational green learning and keep in touch with advanced green technology capabilities, so as to keep the enterprise in a leading position in the turbulent technological environment.
Our findings may be of value to inform policy maker’s efforts in promoting sustainable development goals. The results of this research contribute to providing an approach on how to conduct a better environmental management, which brings more benefits to a better life of the society and the world as a whole. Manufacturing enterprises are the major contributors to environmental damage. The enterprises with green innovation strategy Proactive reduce emissions and start conducting resource efficiency. When most manufacturing enterprises implement this model to solve environmental issues, the whole society will get more benefits on the reduction of environmental degradation, the availability of more green products, the improvements in resource efficiencies and economic development and the betterment of life quality. Policy makers should guide and encourage enterprises to adopt green innovation strategies, and help enterprises to carry out green operation through green subsidies, green financing and exclusive channels for green products.
Point 4: The paper is written in a clear, comprehensive and accessible manner. Nonetheless, some minor language revisions are recommended.
Response4: Thank you for your suggestion. We have polished the language and standardized the grammar and syntactic expression to ensure the academic and normative of this paper.
Once again, we appreciate your insightful suggestions and thank you very much indeed for taking time to read our manuscript and for your detailed review. We appreciate for your time invested, and hope that the correction will meet your standard of approval.
Reviewer 2 Report
Dear authors,
Congratulations for the work presented.
This study advances a conditional process model to understand how green innovation strategy impacts green innovation performance through organizational green learning in a context of green technological turbulence.
The article is well structured, well presented, with clear ideas, accessible language, with a correct analysis and interpretation of the data and with conclusions in line with all the discussion that precedes it and with the objectives initially defined.
The variables under study are properly explored. It is very important, for the present study, that the authors have clearly defined the concepts contained in the keywords, since they are central throughout the entire article (green innovation strategy; organizational green learning; green technology turbulence; green innovation performance).
The hypotheses are synthetically grounded, and in this particular case there may be room for a greater confrontation of theories and studies. Bibliographic references are not abundant, but they are recent and sufficient to support the construction of the hypotheses summarized in Figure 1.
The methodology is aligned with the type of study, is explained in sufficient detail so that the study can be replicated in another context, and is correctly applied.
The results and their discussion are in line with the objectives of the study and correspond to the alignment of the hypotheses raised by the authors.
A more in-depth exercise on the practical implications of this study would strengthen its relevance and interest. So I suggest some more efforts in this direction.
I also suggest that the authors add to the article some clues for future investigations, as well as the limitations of the present study.
Good luck.
Author Response
Response to Reviewer 2 Comments
Reviewer 2
Title: How to Improve Green Innovation Performance: a Conditional Process Analysis (Sustainability-1618595)
This study advances a conditional process model to understand how green innovation strategy impacts green innovation performance through organizational green learning in a context of green technological turbulence. The article is well structured, well presented, with clear ideas, accessible language, with a correct analysis and interpretation of the data and with conclusions in line with all the discussion that precedes it and with the objectives initially defined.
Response: We very much appreciate your constructive comments and thank you for giving us an opportunity to revise and resubmit our study. We believe that your guidance and comments have indeed assisted us significantly in improving our research and its potential contribution. Following your suggestions, we revised the conclusions and added the limitations and future investigations. We believe that we have made our best efforts and provided the optimal solution to your concern. We hope that you will find our ‘response points’ thorough and convincing. In this letter, we include your original comments in black and then set forth how we have dealt with your comments in red in the new, revised manuscript.
Point 1: A more in-depth exercise on the practical implications of this study would strengthen its relevance and interest. So I suggest some more efforts in this direction.
Response 1: Thank you for your suggestion. We agree with your point and have revised the paper in the part of ‘6.conclusions and implications’. We added the section ‘6.2. Implications’ and the specific modifications are as follows:
6.2. Implications
The findings of this study have implications for practitioners as well as some policy implications. Our empirical results show that green innovation strategy has positive influence on green innovation performance. From the practical perspective, this research model provides a more comprehensive understanding for managers of enterprises who want to enhance green innovation performance. Managers who want to get better results from their sustainability initiatives should adopt green innovation strategy to achieve efficient resource usage, CO2 emissions reduction, energy utilization reduction, moderate cost incurred, ethical waste management and natural resources conservation. This result is consistent with a survey on green innovation strategy in Indonesia which has shown that managers need to start with developing a green innovation strategy to improve green performance [68]. Moreover, this study also clearly shows managers that having a strategy is not sufficient enough to directly enhance green innovation performance. Managers need to seek approaches on how to effectively implement green innovation strategy, encourage employees to carry out organizational green learning and keep in touch with advanced green technology capabilities, so as to keep the enterprise in a leading position in the turbulent technological environment.
Our findings may be of value to inform policy maker’s efforts in promoting sustainable development goals. The results of this research contribute to providing an approach on how to conduct a better environmental management, which brings more benefits to a better life of the society and the world as a whole. Manufacturing enterprises are the major contributors to environmental damage. The enterprises with green innovation strategy Proactive reduce emissions and start conducting resource efficiency. When most manufacturing enterprises implement this model to solve environmental issues, the whole society will get more benefits on the reduction of environmental degradation, the availability of more green products, the improvements in resource efficiencies and economic development and the betterment of life quality. Policy makers should guide and encourage enterprises to adopt green innovation strategies, and help enterprises to carry out green operation through green subsidies, green financing and exclusive channels for green products.
Point 2: I also suggest that the authors add to the article some clues for future investigations, as well as the limitations of the present study.
Response 2: Thank you for your suggestion. Following your suggestions, we added ‘6.3 limitations and Future Research’ at the end of the paper. The specific modifications are as follows:
6.3. Limitations and Future Research
Despite the rigor of this study, some limitations need to be considered when interpreting the results and conducting future research. First, this study emphasizes that manufacturing enterprises are more involved in environmental practices. Manufacturing enterprises consume more resources, and the performance brought by green innovation is more obvious than that of other industries. However, enterprises in other industries are also facing the decision of green innovation strategy. Future research may explore the effect of green innovation strategy on green innovation performance in different industry to enrich the research on environmental strategy research. In addition, due to consideration of the availability of data and the workload of this study, this study focuses on the moderating effect of green technology turbulence on the latter half of the model. For further research, our research team will consider the moderating effect of green technology turbulence on the relationship between green innovation strategy and organizational green learning. And finally, in order to investigate the actual situation of enterprises accurately and effectively, our research obtained primary data through questionnaire survey, which may be affected by subjective perception of subjects despite our efforts to eliminate this effect. Therefore, future studies may explore the quantitative measurement of variables to reconfirm the research conclusion of this paper.
Once again, we appreciate your insightful suggestions and thank you very much indeed for taking time to read our manuscript and for your detailed review. We appreciate for your time invested, and hope that the correction will meet your standard of approval.